# Beyond Needle(s) in the Embodied Haystack: Environment, Architecture, and Training Considerations for Long Context Reasoning

**Bosung Kim & Prithviraj Ammanabrolu**
University of California, San Diego
{bosungkim,prithvi}@ucsd.edu

## Abstract

We introduce $\infty$-THOR, a new framework for long-horizon embodied tasks that advances long-context understanding in embodied AI. $\infty$-THOR provides: (1) a generation framework for synthesizing scalable, reproducible, and unlimited long-horizon trajectories; (2) a novel embodied QA task, Needle(s) in the Embodied Haystack, where multiple scattered clues across extended trajectories test agents' long-context reasoning ability; and (3) a long-horizon dataset and benchmark suite featuring complex tasks that span hundreds of environment steps, each paired with ground-truth action sequences. To enable this capability, we explore architectural adaptations, including interleaved Goal-State-Action and Memory-Augmented Goal-State modeling, along with context extension techniques and Context Parallelism, to equip VLM-based agents for extreme long-context reasoning and interaction. Experimental results and analyses highlight the challenges posed by our benchmark and provide insights into training strategies and model behaviors under long-horizon conditions. Our work provides a foundation for the next generation of embodied AI systems capable of robust, long-term reasoning and planning. The datasets and code can be found at pearls-lab.github.io/infini-thor.

## 1 Introduction

Real-world embodied reasoning is a sequential decision-making problem requiring long-horizon planning, where task success depends on both memorizing and reasoning over multiple events that occur far apart in time. Using large pre-trained vision-language-action (VLA) models as policies for such tasks requires surpassing the key challenge of *long-context* reasoning. We seek to answer questions pertaining to what design choices matter in terms of environments, model architectures, and training methods when using VLA models for long-horizon embodied tasks. To this end, we develop a new framework for long-horizon tasks designed to push the boundaries of long-context understanding in embodied AI.

We introduce $\infty$-THOR, a new framework for generation, training, and evaluation of long-horizon embodied tasks. Our benchmark uniquely features tasks with a synthetic final goal, which involves multiple objects that appear at distant time steps, requiring multi-step reasoning across over hundreds of steps. Figure 1 illustrates an example: the agent observes the tomato at an early step (t=17) and the counter top much later (t=560). Then, the final task is given at t=670, which requires the agent to place the tomato on the counter top. This setup highlights the challenge of long-horizon dependency, where key objects and locations must be remembered and acted upon after hundreds of steps. Beyond these long-horizon dependencies, our framework also generates low-level robot-arm manipulation action sequences aligned with the trajectories, enabling agents to bridge from high-level reasoning to fine-grained physical execution.

This long-horizon setup introduces a new challenging task, Needle(s) in the Embodied Haystack (NiEH). Unlike the standard Needle in a Haystack task (Liu et al., 2024), which focuses on recalling a single clue in text, NiEH poses two main challenges: (1) multiple scattered clues (Needles) and (2) multi-modal inputs that combine visual and linguistic observations from the environment (Em-

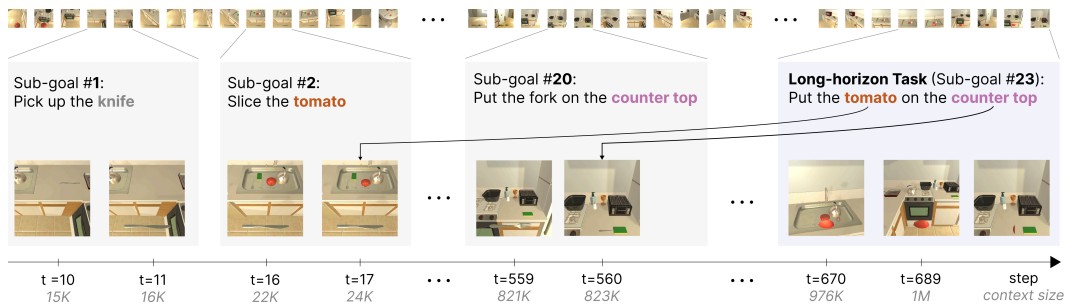

Figure 1: Example of the trajectory and a long-horizon embodied task generated from ∞-THOR. The final goal ("Put the tomato on the counter top" at t=670) requires recalling both the tomato (seen at t=17) and the counter (seen at t=560) to solved the long-horizon task. Context size refers to the input token length when converting the trajectory into the VLM input space.

bodiment). This task is designed to evaluate the agent's ability to recall and reason about previously encountered environmental details, such as identifying objects and recalling performed actions.

Going beyond static evaluations such as NiEH, ∞-THOR also provides an interactive evaluation, allowing agents to execute policies and complete long-horizon tasks within a dynamic environment. To support this, we release a trajectory dataset for training, with episodes over 400 steps in the training set and more than 600 steps in the dev and test sets. These trajectories can be used for imitation learning, and our experiments show that access to longer context during training leads to significant performance gains, highlighting the importance of our dataset for long-context embodied reasoning.

We further investigate various architectural considerations for embodied agents to operate under extreme sequence lengths. We explore two complementary approaches: Interleaved Goal–State–Action model, which directly encodes the full perception–action history as a single multimodal sequence, and Memory-Augmented Goal–State model, which replaces the historical trajectory with text summaries or retrieved visual memories to provide long-term contextual grounding. Moreover, since standard VLMs are constrained by fixed context windows and cannot natively handle inputs exceeding 1M tokens, we evaluate a range of long-context extension techniques, including linear interpolation, dynamic scaling, YaRN, and LongRoPE (Chen et al., 2023; Ding et al., 2024; Peng et al., 2024). Lastly, to support practical training and inference at these extended horizons, we incorporate Context Parallelism via Ring Attention (Liu et al., 2023a), enabling efficient fine-tuning on ultra-long sequences and further improving the model's ability to reason over extended temporal contexts.

We provide comprehensive experiments and analyses, demonstrating both the challenges posed by our benchmark and the behavior of baseline models under long-horizon settings. We investigate a range of training considerations, including different configurations for fine-tuning and long-context adaptation, and evaluate their impact on model performance.

Our contributions are summarized as follows:

- We introduce ∞-THOR, a new framework for generating, training, and evaluating long-horizon embodied tasks, featuring synthetic final goals that require multi-step reasoning across hundreds of steps.

- We propose a novel embodied QA task, Needle(s) in the Embodied Haystack, requiring agents to recall and reason over multiple scattered clues across extended trajectories.

- We release a large-scale trajectory dataset and an interactive evaluation environment to support both offline imitation learning and online policy execution in long-horizon settings.

- We describe architectural adaptations including interleaved Goal-State-Action model and Memory-Augmented model, along with long-context extension and Context Parallelism as tools for handling long-context inputs.

- We present empirical results and analyses, providing insights to the current capabilities and limitations of vision-language-action agents on long-horizon tasks.

## 2 RELATED WORK

Table 1: Comparison of benchmarks. We use Short ($<$ 50 steps), Medium (50-300 steps), and Long ($>$ 300 steps) to describe task horizon, reflecting the approximate number of environment steps required to complete a task in each benchmark. (Inter. w/ env: Interaction with the environment, Mod: Modality, GT: GT actions; single/multi in the QA set column denotes single- and multi-evidence question type. * indicates the number of annotations newly collected in that work.)

| Benchmark / Platform | Task Horizon | Inter. w/ env | Mod | Avg steps | GT | single | multi |
|---|---|---|---|---|---|---|---|
| | | | | Dataset | | QA set | |
| ProcTHOR (Deitke et al., 2022) | ✗ | ✓ | ✗ | ✗ | ✗ | ✗ | ✗ |
| MineDojo (Fan et al., 2022) | Long | ✓ | ✗ | ✗ | ✗ | ✗ | ✗ |
| Habitat 3.0 (Puig et al., 2023) | Long | ✓ | ✗ | ✗ | ✗ | ✗ | ✗ |
| VirtualHome (Puig et al., 2018) | Short | ✓ | multi | 11.6 | ✓ | ✗ | ✗ |
| ALFRED (Shridhar et al., 2020) | Medium | ✓ | multi | 50 | ✓ | ✗ | ✗ |
| ALFWorld (Shridhar et al., 2021) | Medium | ✓ | text | 50 | ✓ | ✗ | ✗ |
| BEHAVIOR-100 (Srivastava et al., 2021) | Med/Long | ✓ | ✗ | ✗ | ✗ | ✗ | ✗ |
| BALROG (Paglieri et al., 2024) | Long | ✗ | ✗ | ✗ | ✗ | ✗ | ✗ |
| EAI (Li et al., 2024b) | Med/Long | ✓ | ✓ | 14.6* | ✓ | ✗ | ✗ |
| EQA (Das et al., 2018) | ✗ | ✗ | ✗ | ✗ | ✗ | ✓ | ✗ |
| MM-EGO (Ye et al., 2025) | ✗ | ✗ | ✗ | ✗ | ✗ | ✓ | ✗ |
| $\infty$-THOR | $\infty$ | ✓ | multi | 627 | ✓ | ✓ | ✓ |

**Long-horizon Planning in Virtual Environments.** AI2THOR (Kolve et al., 2017) provides interactive indoor environments widely used for embodied reasoning research, while ProcTHOR (Deitke et al., 2022) extends these capabilities by procedurally generating scalable environments, potentially facilitating longer trajectories. MineDojo (Fan et al., 2022) offers an open-ended platform within Minecraft, explicitly geared toward tasks requiring extensive long-term planning. Additionally, platforms such as VirtualHome (Puig et al., 2018) and Habitat 3.0 (Puig et al., 2023) have demonstrated suitability for tasks involving long-term interactions and complex activity sequences. However, all of these platforms only provide environments and do not include standardized datasets or benchmark suites to support training and evaluation for long-horizon embodied tasks.

**Embodied QA and Multimodal Needle in the Haystack Tasks.** Embodied QA tasks, such as EmbodiedQA (Das et al., 2018) and MM-EGO (Ye et al., 2025), require agents to answer questions grounded in visual observations with spatial and temporal reasoning, but without active environment interaction during evaluation. Our NiEH task is also related to multimodal Needle in a Haystack (NiH) problems. While early NiH focused on textual recall in long contexts (Liu et al., 2024), recent multimodal extensions add visual inputs (Wang et al., 2024; 2025), though they remain limited to shorter contexts (up to 72K tokens) and lack embodied reasoning or temporal dependencies.

**Datasets and Benchmarks for Long-horizon Embodied Tasks.** Recent efforts have pushed toward long-horizon embodied tasks, where agents must complete multi-step goals with extended temporal dependencies. While ALFRED (Shridhar et al., 2020) and ALFWorld (Shridhar et al., 2021) introduced multi-step instruction-following tasks with action annotations and textual grounding, their task horizons are relatively short, typically under 50 steps. BEHAVIOR-100 (Srivastava et al., 2021) evaluates agent generalization on household activities, some of which require prolonged engagement, but mainly focus on single task. BALROG (Paglieri et al., 2024) is a benchmark for testing the agentic capabilities of long-context LLMs, but its scope is limited to games. Recently, EAI (Li et al., 2024b) proposed a generalized interface to evaluate LLMs for embodied decision making, while our framework focuses on multimodal online evaluation with real-time rewards and a new long-horizon reasoning task (NiEH).

**Long-context Benchmarks.** Outside embodied AI, general benchmarks have addressed challenges in long-context reasoning. Benchmarks, such as LongBench (Bai et al., 2024) and RULER (Hsieh et al., 2024), focus on retrieval or summarization tasks. GSM-$\infty$ (Zhou et al., 2025) extends GSM-8K (Cobbe et al., 2021) to assess mathematical reasoning over extremely long textual inputs. More recently, LMAct (Ruoss et al., 2025) proposed a benchmark for evaluating frontier models' long-context multimodal decision-making on interactive game-based tasks, with up to 1M context lengths.

# 3 ∞-THOR: AN ENVIRONMENT FOR GENERATING, TRAINING, AND EVALUATING LONG-HORIZON EMBODIED TASKS

∞-THOR features with a generation framework for synthesizing long trajectories to train and evaluate AI agents in long-horizon embodied tasks. We build ∞-THOR upon AI2-THOR (Kolve et al., 2017) simulator, an interactive 3D environment for embodied AI research that supports diverse scenes, objects, and agent actions. ∞-THOR enables the creation of unlimited trajectories with arbitrary length, and provides an evaluation setup where agents can interact dynamically with the environment during both training and testing. This supports both offline learning by producing large-scale datasets, and online learning through direct agent-environment interaction.

Each trajectory generated by ∞-THOR consists of multiple task goals, such as "Put a clean sponge on a metal rack" and "Pick up the apple and place it on the microwave", requiring grounded understanding and action to achieve the goal. At the end of each trajectory, the agent is assigned a synthetic long-horizon task that requires reasoning over entities encountered at distant time steps. For the example in Figure 1, the long-horizon task (Sub-goal #23) at step t=689, "Put the tomato on the counter top", depends on observations made far earlier: the tomato at t=17 and the counter top at t=560.

Our generation framework can generate unlimited tasks, the trajectories can be exceptionally long, exceeding 1M context tokens or beyond when the trajectory is processed with LLMs. Moreover, ∞-THOR can generate low-level robot manipulation trajectories which are compatible with the ManipulaTHOR (Ehsani et al., 2021) simulator, supporting both symbolic plans and low-level controls. Successfully completing this task requires the agent to (1) memorize and integrate key environmental information over hundreds of steps, and (2) plan actions based on dependencies that are separated in time, demonstrating the need for long-context reasoning and robust spatio-temporal memory, and (3) execute low-level physical actions through extended manipulation sequences.

## 3.1 STATIC EVALUATION: NEEDLE(S) IN THE EMBODIED HAYSTACK

We first introduce a novel task in the form of a static evaluation: Needle(s) in the Embodied Haystack (NiEH). NiEH is designed to evaluate an agent's ability to recall and reason about environmental states encountered throughout a trajectory. Unlike traditional embodied QA tasks that focus primarily on visual understanding of a single image, NiEH emphasizes reasoning about environmental changes over time, requiring agents to interpret and integrate sequences of multimodal observations.

Figure 2 presents examples of the two NiEH task types. In the single-evidence setting, a question is answerable based on a single observation step; in the multi-evidence setting, multiple temporally distant steps must be combined to answer the question. The NiEH testset includes diverse question types, such as binary ("yes" or "no"), "what"-, "where"-, and "how many"-style questions. These questions span a broad range of difficulty, from simple memory recall (similar to the Needle in a Haystack paradigm) to complex queries that requiring multi-step reasoning across temporally and spatially distributed evidence.

**Testset Construction.** We first replay the generated trajectories and collect the agent's egocentric views, along with all objects that interact with the agent throughout the trajectories, such as objects that are picked up, moved, or even simply observed. Based on these interactions, we apply a set of rule-based templates to generate QA pairs, such as "Q. What object did you slice? A. {object name}" and "Q. Is {obeject name} on the desk? A. Yes/No". Then, we sample questions based on the frequency to ensure diversity across object types, and annotate the GT answer steps using the replay logs.

After generating QA pairs and annotating the GT steps, we cross-validate the answerability of each question with GT images using four different multimodal LLMs: LLaVA-OneVision 7B (Li et al., 2024a), Qwen2.5-VL 7B (Bai et al., 2025), Deepseek-VL 7B (Lu et al., 2024), and Pixtral 12B (Agrawal et al., 2024). Since these models are highly capable at standard visual QA, we filter out the questions that none of the four models successfully answer with GT images. At test time, the entire trajectory is treated as a Haystack and then cropped based on the GT image's depth. Full details on templates, generation rules, and the validation scores of the four models are included in the Appendix.

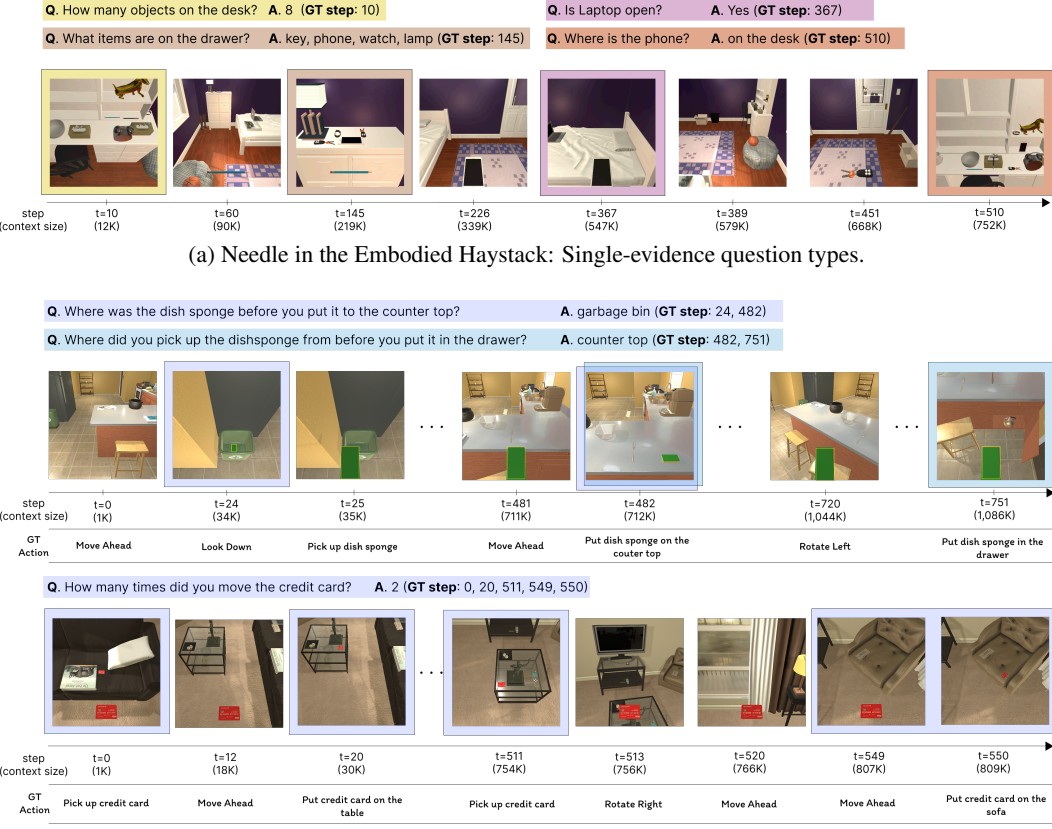

(a) Needle in the Embodied Haystack: Single-evidence question types.

(b) Needles in the Embodied Haystack: Multi-evidence question types.

Figure 2: Example of N(s)iEH task and Ground-truth steps.

**Challenges in Needle(s) in the Embodied Haystack.** The NiEH task introduces two key challenges for current models. First, many questions require reasoning over multiple temporally distant events. As shown in Figure 2(b), the agent moves a `dish sponge` from the garbage bin at $t = 24$, then to the counter top, and later places it into a drawer at $t = 751$. A question such as *"Where was the dish sponge before you put it on the counter top?"* requires the model to recall and chain together multiple actions and locations across hundreds of steps. Second, some questions demand aggregating sparse and temporally scattered evidence from long trajectories. In the second example in Figure 2(b), answering *"How many times did you move the credit card?"* requires the model to track and count all relevant actions occurring from the beginning to the end of the episode. These challenges highlight the need for models that can perform robust long-horizon reasoning across both time and modalities in complex embodied environments.

## 3.2 CONSTRUCTING LONG-HORIZON TRAJECTORIES FOR INTERACTIVE EVALUATIONS

With ∞-THOR's generation framework, we can synthesize long-horizon trajectories to construct training, validation, and test sets for offline learning and evaluation. Our approach builds upon a planner-based method (Kolve et al., 2017), in which we sequentially concatenate multiple single-task demonstrations into a extended trajectory, while maintaining consistency in object states and agent interactions throughout.

To generate each trajectory, we first sample a task type from one of seven predefined templates (e.g., pick two objects and place, pick and place with movable receptacle). We then sample objects that are required to perform the task, such as items to be picked up or receptacles to be interacted with. Based on the sampled task and objects, we use a classical task planner that operates on domains specified in the Planning Domain Definition Language (PDDL) (McDermott et al., 1998) to generate ground-truth action sequences. Only successful rollouts (re-simulated without failure) are retained,

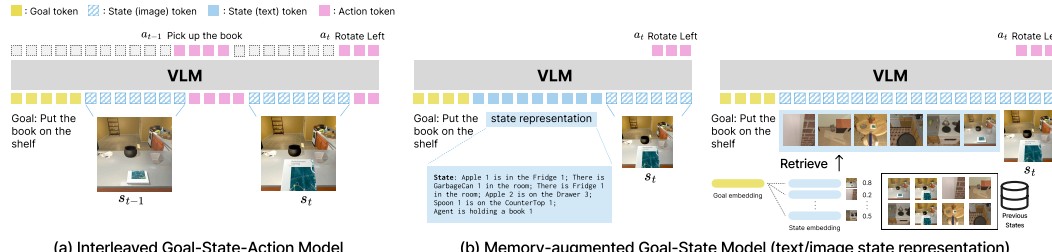

Figure 3: Overview of our Vision-Language-Action architectures.

ensuring the reproducibility and reliability of our trajectories. We then concatenate these successful demonstrations to construct long-horizon sequences that span hundreds of steps.

For the final goal, the involved objects are sampled exclusively from those seen during the early 20% and the final 20% of the trajectory. This enforces a long-term temporal dependency between two objects that must be jointly referenced to complete the final task. Through this procedure, we generate 2,456/125/225 trajectories for the training, validation, and test sets, respectively.

We use a similar approach for low-level manipulation actions. For the Pick-up action, a trajectory is generated from the robot arm's start point to the object's target point (reversed for the Put action). Each

Table 2: Dataset Statistics. "low" denotes low-level robot arm manipulation actions.

| NiEH testset | | Single-clue | Multi-clue |
|---|---|---|---|
| # of question-answer pair | | 829 | 474 |

| Trajectory | Train | Dev | Test |
|---|---|---|---|
| # trajectory | 2,456 | 125 | 225 |
| # avg/max subgoals | 14/30 | 16/24 | 18/33 |
| # avg/max steps | 405/654 | 613/890 | 627/952 |
| # avg token length | 602K | 880K | 912K |
| # max token length | 954K | 1.2M | 1.3M |
| # avg steps (low) | 3,000+ | 5,000+ | 5,000+ |

trajectory is executed through fine-grained arm movements $(\Delta x, \Delta y, \Delta z)$, and only successful rollouts are retained. Details on task types and a pseudo-algorithm for the generation process are provided in the Appendix.

## 4 ARCHITECTURES FOR LONG-HORIZON VISION-LANGUAGE-ACTION MODELS

Long-horizon embodied tasks pose significant challenges due to the need to interpret multimodal inputs (vision, language) and produce coherent sequences of actions over hundreds of interactive steps. However, existing VLA models either use separate encoders for vision, language, and action modules (Shridhar et al., 2020), or focus on short-horizon in constrained environments (Brohan et al., 2023; Kim et al., 2024), where decisions depend only on the most recent observation and a single instruction. While recent multimodal LLMs like LLaVA (Liu et al., 2023b), MiniGPT-4 (Zhu et al., 2023), and Llama 3.2 (AI, 2024) show strong multimodal reasoning abilities, they primarily trained and evaluated on static inputs (e.g., single or few images). This setting is not directly suited to long-horizon embodied tasks, which are image- and interaction-dominant, involving hundreds of egocentric frames and continuous vision-language-action sequences. Moreover, many state-of-the-art proprietary models are only accessible via paid, rate-limited APIs, which makes large-scale, real-time experimentation in embodied settings substantially more difficult and costly in practice.

In this work, we explore two ways of leveraging a VLM as a unified VLA model by utilizing an interleaved input structure of goal, state (visual observations), and action tokens, as illustrated in Figure 3: (1) Interleaved Goal–State–Action Model and (2) Memory-Augmented Goal–State Model.

**Interleaved Goal–State–Action Model** treats the entire perception–action stream as a single unified sequence. At each timestep, the input consists of the task goal followed by all past visual observations and actions in an interleaved order $(g_0, s_0, a_0, \ldots, s_t, a_t)$. This design enables the VLM backbone to jointly reason over multimodal information and autoregressively predict the next action conditioned on the full trajectory history. By leveraging interleaved modeling, our architecture

supports coherent decision-making over temporally distant information while maintaining grounded behavior in dynamic settings.

**Memory-Augmented Goal–State Model** incorporates compact history representations into the input context. Instead of supplying the full trajectory, the model conditions on either (1) textual summaries of past states and environment information (Figure 3-(b)-left), or (2) a retrieved set of relevant historical images selected via embedding similarity (Figure 3-(b)-right). These memory tokens provide distilled representations of long-term context, allowing the model to focus on the current observation while still retaining access to key historical information. By combining memory tokens with the current state, the model can maintain semantic grounding over long horizons without exceeding context-length constraints. This architecture supports scalable long-horizon reasoning and offers a flexible mechanism for integrating multimodal memory into VLA models.

**Context Extension and Context Parallelism**. Given the limitations in context length of most LLMs, using off-the-shelf models is insufficient for processing long inputs such as those exceeding 1M tokens, particularly in our interleaved Goal-State-Action setting. We explore various long-context extension techniques that allows the model to generalize to longer input sequences without retraining from scratch. Specifically, we consider: **Linear Interpolation**(Chen et al., 2023): Rescales input positions to fit within the pretrained RoPE range by linearly interpolating positional indices; **Dynamic Scaling**(Chen et al., 2023): Adapts RoPE frequencies at runtime based on the input sequence length, using a linear rescaling to maintain consistent positional encoding behavior across varying lengths; **YaRN**(Peng et al., 2024): Dynamically interpolates attention frequencies during inference, balancing between pretrained and extrapolated positional regimes; **LongRoPE**(Ding et al., 2024): Augments RoPE with specially designed extrapolation functions, enabling robust generalization to long sequences without degrading attention quality. We apply these techniques during fine-tuning, at inference time, or both.

We also incorporate Context Parallelism built upon Ring Attention (Liu et al., 2023a), which cyclically exchanges key-value shards across devices to compute full attention with significantly reduced memory overhead. We apply these techniques during fine-tuning, inference, or both as essential tools for handling long-context inputs and evaluate how effectively they improve practical performance when interleaved trajectories become substantially long.

# 5 Experiments

## 5.1 Static Evaluation: Needle(s) in the Embodied Haystack

We first evaluate model performance on the Needle in the Embodied Haystack (NiEH) and Needle**s** in the Embodied Haystack (NsiEH) tasks, which test an agent's ability to retrieve and reason over sparse evidence scattered throughout long embodied trajectories.

**Building a Embodied Haystack.** Unlike the traditional Needle in the Haystack setup, which inserts a target sentence into a long text corpus like a book, we use the entire embodied trajectory as the input context. To simulate different reasoning depths, we crop the input sequence either from the beginning or the end based on the GT image's position. In the NsiEH task, where multiple evidences are scattered throughout the trajectory, we fill the context with intermediate steps in between the GT steps keeping their temporal order.

Figure 4 presents the performance of LLaVA-OneVision (OV) 7B (Li et al., 2024a) model across various context extension methods. Linear Interpolation, Dynamic Scaling, and LongRoPE scaling all struggled with very long contexts beyond 128K tokens (the results of Linear Interpolation are excluded from Figure since it fails at all examples). YaRN consistently outperformed other methods, effectively handling contexts above 384K tokens, likely due to its architectural alignment with LLaVA-OV's Qwen2 LM backbone, which employs RoPE and YaRN scaling during pretraining. YaRN performed best at moderate scaling factors (e.g., x4), however, further scaling to x8 and x16 did not yield additional gains. In particular, x16 slightly improved performance in the 256K–384K token range but led to degradation notably at shorter context sizes (<64K), suggesting that excessive scaling may introduce instability and negatively impact performance. Overall, all methods fail beyond 512K tokens, highlighting a need for improved long-context methods.

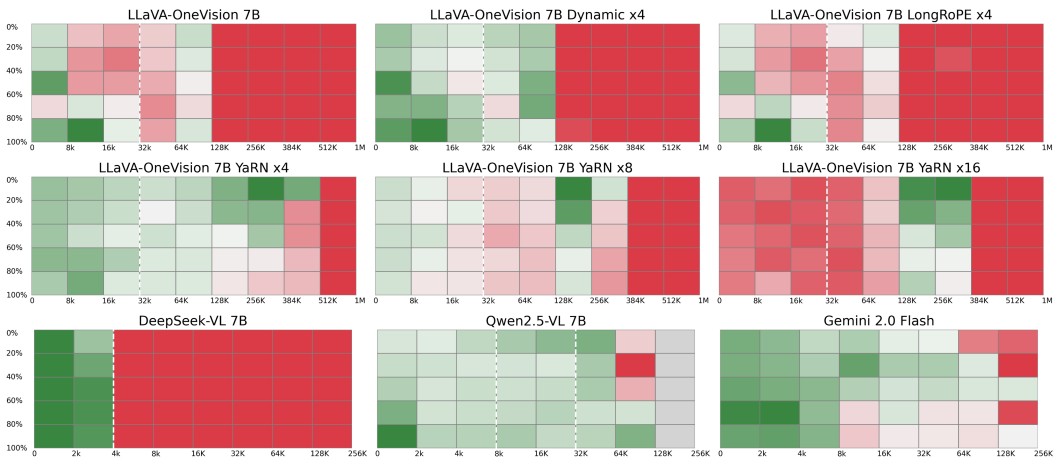

(a) Results of Needle in the Embodied Haystack (NiEH).

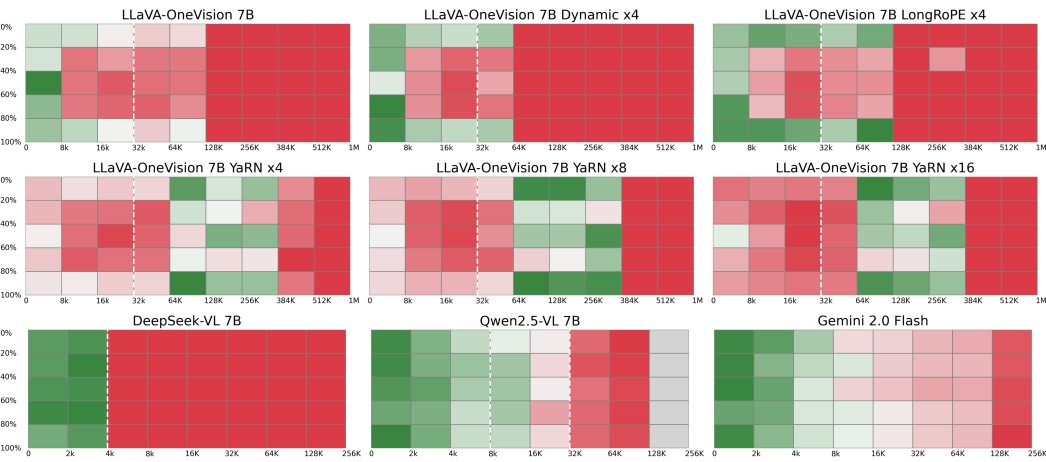

(b) Results of Needles in the Embodied Haystack (NsiEH).

Figure 4: Results of Needle(s) in the Embodied Haystack. The x-axis denotes the input context length (tokens), and the y-axis denotes the needle depth (percentage position of the clue within the trajectory). The white dashed line denotes the maximum pre-training context length of each model. Qwen2.5-VL was pre-trained initially with an 8K token context window and incrementally scaled up to 32K tokens in subsequent stages (Bai et al., 2025). The gray area indicates contexts not applicable (N/A) due to the model's smaller image token size limiting sequences to under 128K tokens. Context Parallelism is applied to all experiments with the context size over 384K.

Figure 4 also includes additional results from DeepSeek-VL 7B (Lu et al., 2024), Qwen2.5-VL 7B (Bai et al., 2025), and Gemini 2.0 Flash[1] (Google DeepMind, 2024). Each model processes images into tokens differently—DeepSeek (576 tokens/image), Qwen2.5 (121 tokens/image), and Gemini 2.0 Flash (258 tokens/image)—which consequently impacts maximum context lengths when transforming N(s)iEH sequences into tokenized inputs (DeepSeek-VL: 512K, Qwen2.5-VL: 128K, Gemini 2.0 Flash: 256K). DeepSeek-VL struggles beyond its 4K pretrained context limit. Qwen2.5-VL maintains stronger performance up to approximately 64K tokens, benefiting from its smaller per-image tokenization, but performance notably declines on NsiEH at longer contexts. Gemini 2.0 Flash performs robustly up to 8K tokens but degrades beyond 128K, especially on NsiEH tasks, highlighting room for improvement in complex long-range multimodal reasoning.

**Architectural Comparison.** Table 3 presents N(s)iEH task performance across different architectures in a setting where full trajectory context is available. LLaVA-OV fails to process these full trajectories (>1M tokens), exposing an inherent limitation in its ability to handle long contexts.

---

[1] We used the `gemini-2.0-flash-001` version for all experiments.

| | Single-clue | | | Multi-clue | | |
|---|---|---|---|---|---|---|
| | LLaVA-OV | Qwen2.5-VL | Gemini-2.0 Flash | LLaVA-OV | Qwen2.5-VL | Gemini-2.0 Flash |
| (a) Full trajectory (Image-only) | 0.0 | 44.67 | 35.44 | 0.0 | 19.34 | 37.08 |
| (b) Interleaved Goal-State-Action | 0.0 | 51.64 | **83.04** | 0.0 | 43.24 | **62.72** |
| (c) Memory-Augmented (Text) | 51.33 | 54.38 | 56.21 | 53.83 | 51.33 | 40.99 |
| (d) Memory-Augmented (Image, Top-10) | 35.80 | 33.99 | 25.18 | 22.15 | 18.63 | 21.18 |
| (e) Memory-Augmented (Image, Top-20) | 37.36 | 42.68 | 28.16 | 23.80 | 21.46 | 30.25 |

Table 3: QA task performance on open-ended questions in the N(s)iEH task across different architectures in a setting where full trajectory context is available. LLaVA-OV model fails when provided with the full trajectory as input (>1M). GPT-4o is used as an LLM judge for evaluation. For Memory-Augmented approaches, we used a CLIP encoder (Radford et al., 2021) to compute embedding similarity between goal descriptions and historical state observations, selecting the Top-10 and Top-20 most relevant images.

While the interleaved Goal-State-Action model provides rich historical context via text and visual states, results indicate that Qwen2.5-VL fails to benefit from this information due to a lack of underlying long-context understanding. In contrast, Gemini 2.0 Flash effectively leverages this temporal context. For Memory-Augmented approaches, textual memory demonstrates significantly higher performance than image-based memory, as textual state representations provide denser information within a shorter context size. Notably, Gemini 2.0 Flash exhibits a notable tendency to hallucinate object locations in image-only settings without language grounding (Table 3-(a), (c), and (d)).

**Single vs Multi-evidence Reasoning.** Comparing NiEH to the more challenging NsiEH task, we observe a significant performance drop in the multi-evidence setting. This is especially pronounced for mid-depth questions involving sparse or distant evidence (e.g., *"Where was the Mug before you put it on the CounterTop?"*) or questions requiring the aggregation of multiple clues (e.g., *"How many times did you move the Apple?"*), as shown in Figure 2(b). These results demonstrate that our NiEH and NsiEH tasks pose a substantial challenge to current long-context models and success requires both fine-grained temporal memory and multi-evidence reasoning across extended interactions.

## 5.2 Interactive Evaluation in ∞-Thor

To measure agent performance on our long-horizon test set, we conduct an interactive online evaluation using the AI2THOR for high-level planning tasks and the ManipulaTHOR for low-level manipulation tasks.

**Plan-level Evaluation.** We evaluate agent performance based on reward accumulation given previous states and actions. Evaluation is performed at the plan-level, where a plan represents a short sequence of actions aimed at a specific intermediate goal. For example, a "Go to location" plan comprises actions like `Move Ahead`, `Rotate Right/Left`, and `Look Up/Down`. Each trajectory is thus composed of multiple sequential plans. At each step, agents are presented with a task goal alongside the previous state-action sequences, predicting subsequent actions and interacting continuously with the environment until the plan completion. Further evaluation details are provided in the Appendix.

**High-level Planning Task.** We fine-tune the LLaVA-OneVision 7B model on our training set while freezing the vision encoder. We used 8 H100 GPUs with both tensor parallelism Shoeybi et al. (2020) and pipeline parallelism Huang et al. (2019) for the 32K context size, while Context Parallelism is employed for training on larger context sizes (64K and 130K). Additional training specifics are available in the Appendix. Figure 5 presents the accumulated rewards over time across six experimental configurations. The Planner trajectory serves as the performance upper bound.

Figure 5(a) compares different context extension methods at a fixed scaling factor of x4. Similar to the NiEH results, YaRN consistently achieves the highest performance showing very close performance to Planner. Figure 5(b) explores YaRN scaling at different scaling factors (x4, x8, and x16). Interestingly, increasing the scaling factor beyond x4 does not significantly improve performance, indicating a diminishing return for larger scaling factors. Figure 5(c) demonstrates the effectiveness of fine-tuning with Context Parallelism, enabling scaling of context lengths up to 64K and 130K tokens. At a 130K context size, the model can learn sequences comprising approximately 86 steps,

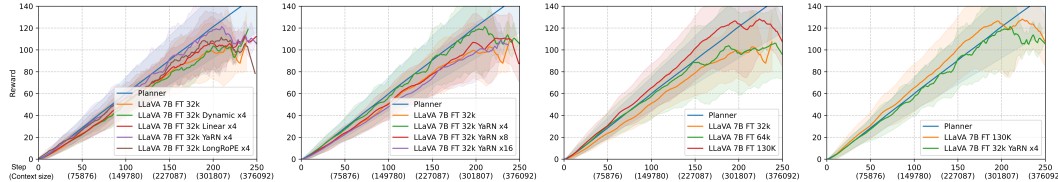

Figure 5: Agent's reward across different experimental configurations for high-level planning tasks. We compare (a) context extension methods at fixed scaling (x4), (b) varying YaRN scaling factors, and (c) fine-tuning with different context lengths using Context Parallelism. (d) summarizes the most effective strategies, highlighting that exposure to longer contexts during training significantly improves performance. Non-planner models cannot generate valid actions after around 250 steps ($\approx$376K in context size). More configurations are in Figure 7 in Appendix.

substantially longer compared to only 22 steps with a 32K context size. This shows that exposure to longer context during training significantly enhances model performance, suggesting that incorporating more long-horizon data by $\infty$-THOR could further improve model capabilities. Additional scaling at evaluation after fine-tuning (Figures 7(d,e) in Appendix) provides no improvement and may even degrade shorter-context performance. Based on these observations, we can conclude that fine-tuning strategies are most effective when long-trajectory datasets are available. In the absence of extensive training data, employing YaRN scaling at x4 yields performance comparable to the Planner upper-bound, particularly within context lengths under 200K tokens (Figure 5(d)).

**Low-level Manipulation Task.** Figure 6 shows agent rewards on low-level manipulation tasks using OpenVLA-7B (Kim et al., 2024) and SpatialVLA-4B (Qu et al., 2025). Ego-centric views are provided as image input, and the models generate the next action $(\Delta x, \Delta y, \Delta z)$ to control the robot arm. Both models perform below the Planner due to differences in robot arm configuration and out-of-distribution inputs, though SpatialVLA achieves slightly higher and more stable rewards than OpenVLA over long sequences. We provide more results and analysis in Appendix C.

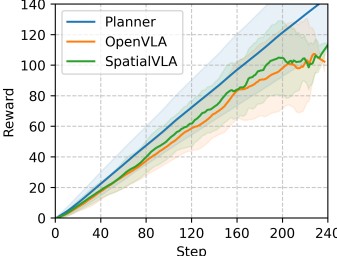

Figure 6: Agent's reward with low-level manipulation tasks. Low-level VLA models are used to control the robot arm for Pick-up and Put actions.

## 6 CONCLUSION

We presented $\infty$-THOR, a new framework for long-horizon embodied tasks designed to advance long-context understanding in embodied AI. Our framework enables scalable synthesis of long, complex trajectories paired with high- and low-level action sequences, and supports both offline training and online interaction with the environment. As part of this framework, we introduced a novel embodied QA benchmark, Needle(s) in the Embodied Haystack, that challenges agents to reason over sparse, temporally distant visual evidence embedded within extended trajectories. To equip models for this setting, we explored architectural adaptations including interleaved Goal–State–Action and Memory-Augmented Goal-State modelings, along with context extension techniques such as YaRN and LongRoPE, and efficient fine-tuning via Context Parallelism. Our experiments demonstrate that exposure to longer contexts during training significantly improves model performance, and the limitation of existing context extension techniques struggle with long-context reasoning. We hope our framework and benchmark encourage further research into models capable of robust long-horizon reasoning under realistic, interactive environments.

## REPRODUCIBILITY STATEMENT

Experiments in this work were conducted using the publicly available simulators AI2-THOR and ManipulaTHOR, which allow for the deterministic replaying of all trajectories provided by $\infty$-THOR. We provide the complete source code, along with the generated datasets and configurations used for each experiment, and this will be made available in a public repository upon publication.

Furthermore, all VLM models used in our experiments are open-source and available on Hugging Face. For the proprietary model Gemini Flash, we have declared the specific version of Gemini Flash used in the paper, and our experiments can be reproduced using the Google Cloud APIs. While the initial generation of trajectories involves some randomness in the sampling of target objects and destination positions, the resulting trajectories themselves are fully reproducible within the simulator environments. The detailed data processing steps and the experimental setup are further described in the Appendix.

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

# A    DATASET CONSTRUCTION

## A.1    BUILDING THE NEEDLE(S) IN THE EMBODIED HAYSTACK BENCHMARK

We construct the Needle(s) in the Embodied Haystack benchmark in three stages: 1) Trajectory Replay and Metadata Collection; 2) Rule-Based QA Generation; and 3) Cross-validation with Multimodal LLMs. The following sections provide detailed descriptions of each step.

### A.1.1    TRAJECTORY REPLAY AND METADATA COLLECTION

We first replay 225 test trajectories generated by $\infty$-THOR, logging both visual observations (agent's egocentric views) and structured metadata at each timestep. For every step, we store the list of visible objects, agent-inventory items, openable containers, and their contents from the simulator. This produces a fine-grained interaction log that captures grounded scene dynamics over time.

An example of the collected metadata at a single timestep is shown below:

```
Example of metadata entry

{
    "img_idx": 2,
    "img_filename": "000000002.png",
    "step": 1,
    "object_log": {
        "visible": ["Shelf", "Vase", "Book"],
        "pickupable": ["Vase", "Book"],
        "isOpen": [],
        "inven_obj": [],
        "receptacles": ["Shelf"],
        "recep_objs": {
            "Shelf": ["Vase", "Book"]
        }
    }
}
```

Each metadata entry corresponds to a low-level action step and provides the semantic state of the scene, enabling the construction of temporally grounded QA instances in later stages.

### A.1.2    RULE-BASED QA GENERATION

To construct the QA set, we apply rule-based generation templates to each trajectory using its sequence of low-level actions and associated metadata. The QA generation process involves parsing the agent's interactions with objects, containers, and the environment, and applying a set of hand-crafted rules to synthesize grounded questions.

Our QA generation logic covers a diverse range of question types, including object presence, object state, location tracking, slicing actions, container content reasoning, and action counting. For instance, if an object is seen for the first time at a particular step, a presence question such as "Is there any `apple` in this room?" is generated. Similarly, after a `PutObject` action, location-based questions like "Where was the `apple` before you put it to the `microwave`?" are produced. When slicing actions happen, we create questions about the object being sliced and other nearby items (e.g., "What objects were in the `Fridge` when you sliced the `apple`?"). Then, we sample questions based on the frequency to ensure diversity across object types, and annotate the GT answer steps using the replay logs. Table 4 summarizes the types of questions generated, and corresponding trigger conditions and example templates.

### A.1.3    CROSS-VALIDATION WITH MULTIMODAL LLMS

To ensure the answerability and clarity of the generated QA pairs, we perform cross-validation using four powerful multimodal LLMs: LLaVA-OneVision 7B (Li et al., 2024a), Qwen2.5-VL 7B (Bai et al., 2025), Deepseek-VL 7B (Lu et al., 2024), and Pixtral 12B (Agrawal et al., 2024). Each model

Table 4: QA types, trigger conditions, and corresponding question templates used in rule-based generation.

| QA Type | Trigger Condition | Example Template(s) |
|---|---|---|
| object presence (Yes/No) | object appears visibly in the trajectory | `Is there any {obj} in this room?` `Have you seen a/an {obj}?` |
| open state questions | container marked as open in metadata | `Was {container} open?` |
| object location tracing | sequences of `Pickup` and `PutObject` actions | `Where was {obj} before you put it to {container}?` `Where did you move the {obj} from the {container}?` `Where is {obj} now?` |
| slicing-based questions | `SliceObject` action detected in trajectory | `What did you slice?` `What objects were in/on the {container} when you slice the {obj}?` |
| container content | container visibility with non-empty contents | `What objects were in/on the {container}?` `What object did you put in/on the {container}?` |
| put action questions | unique `PutObject` action for a container | `What object did you put in/on the {container}?` |
| final object state | final location of an object at episode end | `Is {obj} in/on the {container}?` `What objects are in/on the {container}?` `How many objects were in/on the {container}?` |
| movement counting | object picked up more than once | `How many times did you move {obj}?` |

is prompted with the GT images corresponding to the annotated QA steps and asked to answer the associated questions. Given their strong performance on standard visual QA tasks, we use these models to assess whether a question can be correctly answered or not. We keep only the QA pairs that are correctly answered by at least one of the four models, and discard those that fail across all models. This helps improve dataset quality and filtering out ambiguous or visually ungroundable questions. Table 5 shows the accuracy of each model on the finalized QA set when evaluated with GT images. Notably, even with access to GT images, all models struggle with questions requiring reasoning over three or more evidence steps. To maintain the benchmark's difficulty and support evaluation of more capable models in future, we manually inspect the multi-clue questions and include those that are answerable.

Table 5: QA accuracy (%) of multimodal LLMs on ground-truth images.

| Model | Size | # of clues (GT steps) | | | Total |
|---|---|---|---|---|---|
| | | 1 | 2 | ≥3 | |
| LLaVA-OneVision | 7B | 86.61 | 68.55 | 23.74 | 71.15 |
| Qwen2.5-VL | 7B | 85.94 | 89.83 | 64.40 | 82.20 |
| Deepseek-VL | 7B | 81.56 | 39.14 | 22.57 | 62.88 |
| Pixtral | 12B | 91.34 | 39.60 | 58.56 | 76.25 |

## A.2 CONSTRUCTING LONG-HORIZON TRAJECTORIES

To synthesize long-horizon trajectories, we construct each trajectory by sequentially chaining successful sub-tasks sampled from a predefined set of task templates. This process is illustrated in Algorithm 1. We begin by sampling a task template from a fixed task pool, which includes goal types such as `pick and place simple`, `pick two obj and place`, and `pick and place with movable recep`. Each sampled template requires relevant objects in the scene (e.g., pickupable items, target receptacles), which are then used to define the goal for that task.

We use a classical task planner, which operates over PDDL-defined domains (Shridhar et al., 2020), to generate a low-level action sequence for the sampled goal, and simulate this plan in an interactive environment. If the rollout fails (e.g., due to collisions, object occlusions, or unreachable conditions), we discard the sequence and re-sample from the task pool. Otherwise, the successful rollout is retained and appended to the ongoing trajectory.

This sampling-execution loop is repeated until a long trajectory with a desired number of sub-goals is formed. The resulting synthetic long-horizon trajectory consists of multiple sub-goals concatenated into a continuous sequence. To induce long-term temporal dependencies, the final sub-task is constrained to involve only objects that appear in the early 20% and late 20% of the overall trajectory, requiring the agent to integrate temporally distant evidence to answer associated questions.

---

**Algorithm 1** Construct Long-horizon Trajectory

---

1: **Input:** Task Pool $\mathcal{T}$, max sub goals $N$
2: **Output:** Long-horizon trajectory $\tau$
3: Initialize empty trajectory $\tau \leftarrow []$
4: **while** $\text{len}(\tau) < N$ **do**
5:     Sample task template $g \sim \mathcal{T}$ and objects
6:     Plan action sequence $\pi_g$ by planner
7:     **if** $\text{Simulate}(\pi_g)$ is successful **then**
8:         Append $\pi_g$ to trajectory: $\tau \leftarrow \tau \parallel \pi_g$
9:     **else**
10:         Discard and re-sample
11:     **end if**
12: **end while**
13: // Final sub-task with long-term object dependency
14: Sample $g_{\text{final}} \sim \mathcal{T}$ and objects in early 20% and late 20%
15: Plan and simulate $\pi_{\text{final}}$ using restricted objects
16: **if** $\text{Simulate}(\pi_{\text{final}})$ is successful **then**
17:     Append $\pi_{\text{final}}$ to trajectory: $\tau \leftarrow \tau \parallel \pi_{\text{final}}$
18: **else**
19:     Repeat sampling until success
20: **end if**
21: **return** $\tau$

---

# B   TRAINING AND EVALUATION DETAILS

## B.1   INTERACTIVE EVALUATION IN ∞-THOR

**Training.** We fine-tune the LLaVA-OneVision 7B model on our training set while freezing the vision encoder. The model is trained using a next-action prediction objective, where only the action tokens are optimized, conditioned on the goal and state tokens. Table 6 summarizes the training specifications for different context lengths. For 32K training, we apply tensor parallelism with a degree of 4 and pipeline parallelism with a degree of 2, utilizing 8 H100 GPUs in total. Since pipeline parallelism requires the batch size to match the pipeline degree, we set the batch size to 2. For longer context lengths, we use context parallelism: 8-way for 64K (on 8 GPUs) and 16-way for 130K (on 16 GPUs). All models are fine-tuned for approximately 3 epochs with a learning rate of 1e-5, using the AdamW optimizer and a linear learning rate schedule with a 0.03 warmup ratio.

Table 6: Training specifications for different context lengths.

| Context Length | Parallelism | # GPUs | Training Time |
|:---:|:---:|:---:|:---:|
| 32K | Tensor (4) + Pipeline (2) | 8 | 160 hrs |
| 64K | Context (8) | 8 | 120 hrs |
| 130K | Context (16) | 16 | 134 hrs |

**Plan-Level Evaluation.** We evaluate agent performance using a plan-level framework, where each plan corresponds to a short sequence of actions aimed at achieving a specific intermediate sub-goal (e.g., navigating to an object, placing an item). A trajectory is composed of multiple such plans, executed sequentially. For the interactive evaluation, the agent is presented with the current plan's goal along with the history of previous GT states and actions. Using this context, the agent predicts the next action and interacts step-by-step with the environment. The interaction continues until the current plan is either successfully completed or terminated due to failure (e.g., collisions or deadlocks). After each plan, the context is reset to include the GT actions and states from the completed portion of the trajectory, and the agent proceeds to the next plan. This ensures that each plan is evaluated independently, conditioned only on the correct prior history. The agent's performance is measured via cumulative reward across all plans in the trajectory. Pseudocode for this evaluation procedure is provided in Algorithm 2.

---

**Algorithm 2** Plan-Level Evaluation

1: **Input:** Trajectory $T = \{P_1, P_2, \ldots, P_N\}$, Agent policy $\pi$, Environment $\mathcal{E}$
2: **Initialize:** Reward $R \leftarrow 0$
3: Initialize state and history with initial observation
4: **for** each plan $P_i$ in $T$ **do**
5:     Initialize context with GT actions up to $P_{i-1}$
6:     **while** not *done* and not *failure* **do**
7:         $a_t \leftarrow \pi(\text{context})$
8:         $s_{t+1}, r_t, done, failure \leftarrow \mathcal{E}.\text{step}(a_t)$
9:         Append $(a_t, s_{t+1})$ to context
10:        $R \leftarrow R + r_t$
11:     **end while**
12:     **if** *failure* **then**
13:         Break evaluation
14:     **end if**
15: **end for**
16: **return** Total accumulated reward $R$

---

## C ADDITIONAL RESULTS

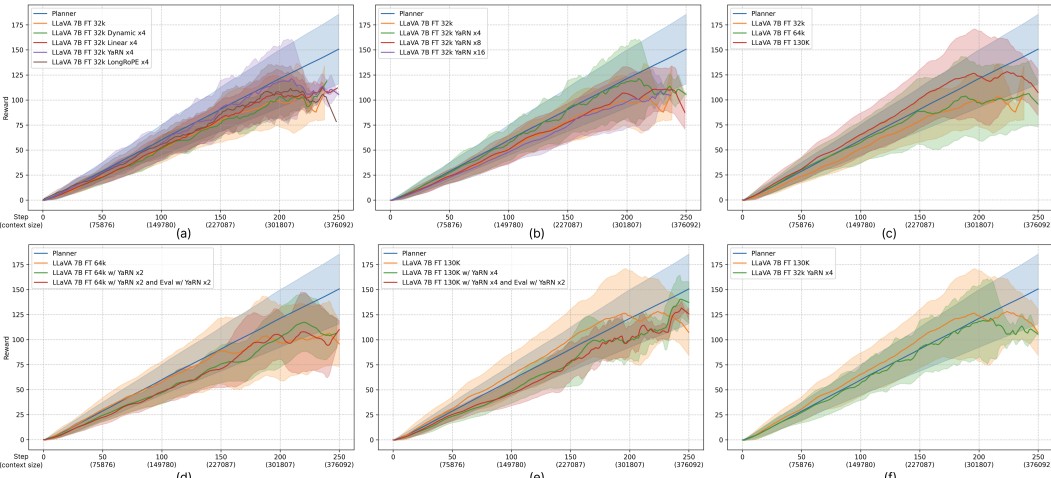

Figure 7: Agent's reward across different experimental configurations: (a) context extension methods at fixed scaling (x4), (b) varying YaRN scaling factors, (c) fine-tuning with different context lengths using Context Parallelism, (d-e) combinations of scaling during both training and inference, and (f) summary of the most effective strategies.

## C.1 INTERACTIVE EVALUATION: HIGH-LEVEL PLANNING

**Results and Discussion.** Figure 7 presents the accumulated rewards over time across six experimental configurations. The Planner trajectory represents the performance upper bound. Our analysis focuses on addressing the following key questions:

*Q. Which context extension methods perform best?* Figure 7(a) compares different context extension methods at a fixed scaling factor of x4. Similar to the NiEH results, YaRN consistently achieves the highest performance showing very close performance to Planner.

*Q. Does further scaling enhance performance?* Figure 7(b) explores YaRN scaling at different scaling factors (x4, x8, and x16). Interestingly, increasing the scaling factor beyond x4 does not significantly improve performance, indicating a diminishing return for larger scaling factors.

*Q. Is fine-tuning on a dataset with long trajectories effective?* Figure 7(c) demonstrates the effectiveness of fine-tuning with Context Parallelism, enabling scaling of context lengths up to 64K and 130K tokens. At a 130K context size, the model can learn sequences comprising approximately 86 steps, substantially longer compared to only 22 steps with a 32K context size. This shows that exposure to longer context during training significantly enhances model performance, suggesting that incorporating more long-horizon data by ∞-THOR could further improve model capabilities. We note that context extension methods were not applied in this experiment.

*Q. Does combining context extension methods during both training and inference provide additional benefits?* Results of experiments with scaling at both training and evaluation (Figures 7(d) and 7(e) in Appendix) indicate that additional scaling at evaluation after fine-tuning with scaled RoPE provides no further performance improvement and may degrade performance at shorter context lengths (≤300K tokens).

Based on these observations, we can conclude that fine-tuning strategies are most effective when long-trajectory datasets are available. In the absence of extensive training data, employing YaRN scaling at x4 yields performance comparable to the Planner upper-bound, particularly within context lengths under 200K tokens (Figure 7(f)).

## C.2 INTERACTIVE EVALUATION: LOW-LEVEL MANIPULATION

Table 7: Success rates of OpenVLA-7B and SpatialVLA-4B on low-level Pick-up and Put tasks in ManipulaTHOR.

| Task | OpenVLA-7B | SpatialVLA-4B |
|---|---|---|
| Pick-up | 16.52% | 18.70% |
| Put | 2.08% | 1.19% |

We evaluate the ability of existing VLA models, OpenVLA-7B (Kim et al., 2024) and SpatialVLA-4B (Qu et al., 2025), to control robot arms on low-level manipulation tasks. The evaluation protocol follows the procedure described in Appendix B.1, with the only difference being that low-level VLA models are used to directly execute Pick-up and Put actions through arm control. For the text input, we provide task-specific instructions ("Pick-up" or "Put" object), since the existing models are only trained on single-task formulations. For the image input, we use an ego-centric camera view. Since these VLA models were originally trained on datasets with different viewing angles, there remains considerable room for improvement by incorporating additional camera perspectives during training and evaluation.

For the evaluation criterion, ManipulaTHOR implements a "magnet sphere" hand mechanism: if a pickupable object is within a specified radius of the agent's hand when the `PickupObject` action is called, the object is successfully picked up. Since neither OpenVLA nor SpatialVLA was trained in the AI2-THOR environment, we relax this success threshold by setting the radius to 0.4 to account for discrepancies between training and evaluation environments. This loosened criterion ensures that small deviations in arm trajectories do not result in an immediate failure, thereby providing a fairer comparison of the models.

Figure 6 presents accumulated rewards on low-level manipulation tasks. Both models underperform relative to the Planner baseline, largely due to differences in robot arm configuration and the out-of-distribution nature of the visual inputs. The results show that SpatialVLA achieves slightly higher and more stable rewards than OpenVLA across long sequences.

Table 7 reports task success rates. Overall performance remains low, with both models struggling on manipulation tasks. We attribute these results primarily to two factors: (1) ManipulaTHOR's egocentric view, which provides nearly identical images for fine-grained arm movements $(\Delta x, \Delta y, \Delta z)$ smaller than 0.01, making it difficult for VLA models to perceive subtle adjustments; and (2) discrepancies between the robot arm in ManipulaTHOR and those assumed by the VLA models, causing many predicted actions to fail in the physical simulation. We expect that fine-tuning on our dataset could provide an effective solution to bridge these gaps.

## D  LIMITATIONS

While $\infty$-THOR enables the generation of arbitrarily long trajectories, the diversity of environment layouts in AI2-THOR is inherently limited. This can lead to repetitive agent behaviors within certain scenes. For instance, compared to kitchen and living room scenes, bedroom scenes tend to involve fewer action types, mostly constrained to simple pick-and-place tasks within small spatial areas. Due to the limited scene size and low task diversity, we excluded bathroom scenes from our dev and test sets. In future work, we plan to integrate $\infty$-THOR with ProcTHOR (Deitke et al., 2022), which supports procedurally generated environments, enabling a broader range of dynamic and diverse scene configurations.

Additionally, although the GT action sequences generated by the PDDL-based planner are sufficient for task completion, they are not guaranteed to be optimal. This may lead to agents learning suboptimal behaviors when trained solely on these demonstrations. Incorporating learning from exploration or reinforcement-based optimization could improve policy quality.

Finally, inference with LLMs in long-context settings remains a computational bottleneck, particularly as context lengths approach 1M tokens. Since our long-horizon tasks require access to information spread throughout the entire trajectory, full-context inference becomes increasingly expensive, even with Context Parallelism. One promising direction is to equip agents with memory systems that selectively keep relevant information from prior steps, allowing the model to reason without reprocessing the full context at each step. Other architectural solutions, such as sparse attention mechanisms (Han et al., 2024; Jiang et al., 2024) or state-space models with linear-time inference (Gu & Dao, 2024), also hold potential for scalable long-context reasoning.

