# OpenReview forum: "Beyond Needle(s) in the Embodied Haystack: Environment, Architecture, and Training Considerations for Long Context Reasoning"
_ICLR.cc/2026/Conference — Submitted to ICLR 2026_

### Official Review · Reviewer_rSXS · 2025-10-28

**Soundness:** 2
**Presentation:** 3
**Contribution:** 2
**Rating:** 4
**Confidence:** 3

**Summary:**

This paper presents a new framework for generating, training, and evaluating long-horizon embodied reasoning tasks. It introduces the Needle(s) in the Embodied Haystack (NiEH) benchmark, which tests agents’ ability to recall and reason over multiple scattered clues across hundreds of environment steps. The framework supports scalable trajectory synthesis using AI2-THOR and integrates architectural techniques—such as interleaved Goal-State-Action modeling, context extension, and Context Parallelism, to enhance long-context reasoning in embodied agents. Extensive experiments reveal the severe performance degradation (“memory cliff”) that occurs as context length increases and analyze how different extension methods mitigate this effect.

**Strengths:**

1. Novel benchmark and task design: NiEH introduces an original embodied QA setting requiring multi-clue, multi-step reasoning over long trajectories, a valuable contribution to evaluating memory and reasoning.
2. Scalable trajectory generation pipeline: The framework builds a reproducible, large-scale dataset based on AI2-THOR, enabling consistent generation of trajectories with hundreds of steps and millions of tokens.
3. Systematic empirical analysis: The experiments provide comprehensive comparisons of context-extension strategies and clearly expose the “memory cliff” phenomenon in existing models.
4. Technical depth: The architectural adaptations (Goal-State-Action modeling, Context Parallelism) are conceptually well-motivated and technically detailed.

**Weaknesses:**

1. Dataset construction methodology: The long trajectories appear to be concatenations of short demonstrations; it remains unclear how object states and interaction continuity are maintained. Why not synthesize single coherent long trajectories directly?
2. Benchmark novelty and scope: The NiEH benchmark closely resembles long-video understanding settings; the distinction between this work and prior long-video benchmarks should be made clearer beyond simply being “embodied.”
3. Limited model evaluation: Most experiments rely on off-the-shelf VLMs with minimal low-level VLA validation. This weakens claims about the framework’s utility for training or improving embodied agents.
4. Visualization clarity: Figure 4 heatmap presentation is visually rich but lacks clear quantitative comparisons, statistical significance, and explicit axis labeling. More intuitive ablations would improve interpretability.
5. Benchmark discrimination: The results highlight engineering aspects of long-context handling but do not clearly show whether the proposed benchmark can distinguish different models’ reasoning abilities in a consistent way.
6. Missing supplementary materials: Several details are deferred to the appendix, but no appendix appears in the submission, limiting reproducibility and clarity.

**Questions:**

The paper provides a valuable benchmark and framework for studying long-context reasoning in embodied AI, with strong engineering and analytic depth. However, the novelty relative to existing long-video settings, limited evaluation coverage, and missing appendices reduce its impact. Strengthening empirical breadth and clarifying benchmark construction would elevate it to a solid accept. Please refer to the weakness above.

---

> ### Author Response · Authors · 2025-12-02
>
> We thank you for the valuable and thoughtful feedback. We address the raised limitations below:
>
> 1. We agree that the trajectory construction process needs to be explained more clearly. One of the core engineering contributions of our framework is to maintain consistent object states and interaction continuity when concatenating multiple sub-tasks. When we “concatenate” demonstrations, we do not simply stitch together independent clips: we replay the planner-generated actions in AI2-THOR so that object states (positions, open/closed, toggled, picked/placed, etc.) evolve continuously across sub-tasks, as if they belonged to a single long, coherent episode. This required non-trivial engineering to ensure that state transitions remain valid and reproducible over hundreds of steps, and we hope this can be regarded as a core contribution of our work.
>
> 2. We appreciate the concern about overlap with long-video understanding. Conceptually, NiEH is indeed related in that it requires retrieving and combining scattered clues across a long temporal span. The key distinctions we aim to emphasize (and will clarify more explicitly) are that:
> * NiEH operates in an embodied setting, where each frame is the result of actions in a 3D environment (not a passively observed video);
> * the trajectories are paired with ground-truth action sequences and sub-goal structure, enabling both offline QA and online experiments within the same framework; and
> * NiEH is coupled with the ability to generate new, arbitrarily long trajectories, which is less common in purely video-based benchmarks.
>
> 3. We agree that broader model coverage would further strengthen the empirical story. In addition to off-the-shelf VLMs, we do train models on our generated trajectories and evaluate them, including in the online setting (Section 5.2), where we show that fine-tuning on our long trajectories yields the best long-context performance.  The choice to include off-the-shelf models was driven in part by fairness of comparison to Gemini, which cannot be fine-tuned.
>
> 4. We appreciate the suggestion to improve the clarity of Figures. We have clarified what axis labels denote in caption in the updated version.
>
> 5. Our primary goal with NiEH is to probe long-horizon reasoning over trajectories, i.e., whether models can correctly answer questions that depend on multiple, temporally distant clues, rather than only exposing generic context-length limitations. We agree that this distinction should be made more explicit. However, we want to note that we already compare multiple models and observe consistent differences in how they handle scattered evidence under identical long-context conditions in Section 5.1.
>
> 6. We apologize for the confusion. We’ve updated the new version with appendix
>
> We thank the reviewer again for the constructive feedback and believe these clarifications and revisions will substantially improve the paper.

---

### Official Review · Reviewer_fHk7 · 2025-10-28

**Soundness:** 2
**Presentation:** 2
**Contribution:** 2
**Rating:** 4
**Confidence:** 3

**Summary:**

The paper introduces a method for generating long multi-goal trajectories in the AI2-THOR environment, along with question-answer pairs whose answers depend on either a single (NiaH) or multiple (NsiaH) past events. This setup requires long-term memory capabilities and enables the evaluation of vision–language question answering over extended sequences.
The generated data is used for the evaluation of VLA / VLM models in long context settings.

In addition, the paper discusses an online interaction VLA design / implementation for such long context settings.

The paper presents empirical evidence showing that several current VLMs struggle in long-context settings where the context length exceeds that used during training / FT.
Furthermore, in another set of experiments, the proposed online VLA design was fine-tuned on the proposed dataset under several configurations and the models were evaluated in an "interactive" setup.

**Strengths:**

- The long context and multi goal trajectory generation method and question-answer pairs generation method are novel and could contribute to the VLA/VLM community.
- The empirical results reflect the poor performance of current methods in long-context settings.

**Weaknesses:**

`W1`: Overall, the paper is not easy to follow, and the presentation lacks clarity in explaining what was actually done. Significant effort is required to understand precisely the main contributions and methodology. The writing would benefit from a more direct, transparent exposition of the key ideas and experimental details.




`W2`: Ultimately, the results in Figure 4 suggest that current architectures (incl. long context solutions) struggle with contexts longer than the training / FT context length. This observation is not particularly surprising.



`W3`: The experimental design of the interactive (online) experiment setup may be flawed. More information is needed to determine with certainty, see `Q5, Q6` below for questions. This could impact the validity of the corresponding conclusions.

My concern is that the context is reset after each sub-goal based on the data (states and actions) of the trajectory from the dataset (generated by the same planner used in the training set), rather than the states and actions produced through the online interaction. In such a case, it is possible that performance are maintained due to overfitting the training set (which was used for the fine-tuning). It is also unclear whether states and actions from previous tasks are relevant for the success on the current task.

Furthermore, such a setting is not a true online evaluation, but rather evaluates each task independently until first failure.



`W4`: line 269: the acronym "PDDL" was never introduced. Please also provide a reference.

`W5`:
> "$\infty$-THOR enables the creation of unlimited trajectories ***with arbitrarily long***, and provides" (line 163)

This sentence is truncated?


`W6`: The paper claims
> We show that interleaved Goal-State-Action modeling ...  is the most practical approach for this class of problems (lines 77-78)

This claim is not supported by the evidence presented in the paper. To show that an approach is "the most practical", one must defined precisely what "practical" means and compare to all relevant existing baselines.


`W7`: The term "embodied AI" is much broader than language-vision based models, and includes non-language models as well (e.g., deep RL). The paper proposes a framework aimed specifically at language-vision driven methods. The scope of the discussion should be clearer and more accurate.

The claim
> "We present empirical results and analyses, providing insights to the current capabilities and
limitations ***of embodied AI systems*** on long-horizon tasks."

suggests a wider scope than what the paper includes. The scope of the claim should be adjusted accordingly.



`W8`:
> ... lack the dynamic interactivity and memory needed for long-horizon embodied tasks involving continuous vision-language-action sequence (line 303)

This statement lacks supporting evidence (either provide empirical evidence or refer to prior works that include such supporting evidence).



`W9`:
> Moreover, many state-of-the-art models are only accessible via proprietary APIs, making them impractical for real-time, controllable embodied settings and managing long-term memory states. (line 305)

The fact that a model is proprietary does not mean that it lacks capabilities. It is unclear what argument this statement aims to support.


`W10`: The plots in Figure 5 are too small. In addition, information about the shaded area is missing.


`W11`: Figure 4 is missing axes labels and a color bar. In addition, tick labels are too small.




`W12`: The setup in Section 5.1 is not clear enough. Are the models fixed? Were they fine-tuned? What data exactly was used for the evaluation? test set only? all dataset splits?


`W13`:
> Our experiments demonstrate that exposure to longer contexts during training significantly improves model performance (line 482)

This observation is inline with existing literature and is not surprising or new.


`W14`: The literature review pertaining to the method discussed in Section 4 is insufficient. Specifically, regarding the "goal-state-action" design, the idea of concatenating tokens of various modalities along the temporal axis was studied in many prior works, see [1][2][3] for example.

This "goal-state-action" approach can not be considered as novel.

Ultimately, in this context, the paper explores fine-tuning such methods to longer context lengths (e.g. 200K+), and shows that long-context performance improve with long-context specific training, which is not surprising.



[1] Reed, S., Zolna, K., Parisotto, E., Colmenarejo, S. G., Novikov, A., Barth-Maron, G., ... & De Freitas, N. (2022). A generalist agent. arXiv preprint arXiv:2205.06175.

[2] Kim, M. J., Pertsch, K., Karamcheti, S., Xiao, T., Balakrishna, A., Nair, S., ... & Finn, C. (2024). Openvla: An open-source vision-language-action model. arXiv preprint arXiv:2406.09246.

[3] Driess, D., Xia, F., Sajjadi, M. S., Lynch, C., Chowdhery, A., Wahid, A., ... & Florence, P. (2023). Palm-e: An embodied multimodal language model.



`W15`: Based on the information in the paper alone, it is unclear how the online interactive evaluation is performed. The corresponding information from the appendix should be included in the main text, or at least a short description of it.

**Questions:**

`Q1`: The term “reasoning” has gained widespread use in recent years, yet there remains no formal definition or consensus on its meaning. What exactly do *you* mean by "reasoning"? Please be precise. This term was used extensively throughout the paper, and its meaning seems to differ based on the context.


`Q2`:
> ... a new framework for generation, training, and evaluation of long-horizon embodied tasks (line 39)

"training / evaluation of [...] tasks", what does it mean to train a task? The paper does not describe training of tasks.


`Q3`: Why is the supplementary material in a separate file?


`Q4`:
> We release a large-scale trajectory dataset and an interactive evaluation environment ... (line 96)

Where is this interactive evaluation environment described?


`Q5`: In the online interactive evaluation, it is stated that
> After each plan, the context is reset to include the GT actions and states from the completed portion of the trajectory, ... (line 871)

what does "GT actions and states" mean in this context? pre-generated actions and states generated with the planner or the states and actions produced during the online interaction with the environment? are there actions and states that are not GT?

If the meaning of GT here is pre-generated planner actions, why do you use these? do you also reset the environment state accordingly? is there a discrepancy between the environment state and the context?

Also, in such a case, the model is effectively evaluated on each sub-goal independently, while terminating upon the first failure, as the context at each sub-task is being reset to a trajectory prefix generated by the (same) planner, used in the training set.


`Q6`: Is there a dependency between sub-goal/tasks? i.e., is it necessary to use information from previous tasks (through the states and actions in the context) to successfully perform the current task (goal)?


`Q7`: Regarding the trajectories generation process: How do you set up the initial environment state? how do you determine the configuration of the elements in the scene (placements, object types, number of objects, etc)? Given an initial setup, how many possible combinations for sub-goals are there?

I am concerned about the overall diversity of the dataset and how it affects the results and conclusions. Based on the information in the paper alone, it is impossible to get a sense of the true diversity of the dataset (I assume that this is in large implied by the AI2-THOR environment). Can you provide further information in this regard?

---

> ### Author Response · Authors · 2025-12-02
>
> We thank you for the valuable and thoughtful feedback. We address the raised limitations below:
>
> W1. We appreciate this feedback and clarified that our primary goal is to introduce a framework for long-horizon embodied tasks that enables systematic studies of architectures, context handling, and training strategies; we also tightened the narrative to present these dimensions more linearly.
>
> W2. While it is expected that models struggle beyond their training context length, our experiments show that (1) the choice and pre-training alignment of context-extension methods strongly affects behavior, (2) simply scaling context length (e.g., ×8, ×16) does not yield proportional gains and can hurt performance, and (3) architectures degrade in qualitatively different ways. We now highlight these findings more clearly.
>
> W3, W15, Q4, Q5. Regarding online interactive evaluation design and GT actions:
>
> We clarified the online evaluation protocol in the main text and appendix: tasks are decomposed into sub-goals, the model interacts until each sub-goal succeeds or fails, and after each sub-goal we reset to the ground-truth prefix and proceed
> We reset to GT prefixes to stress-test how well models use long multimodal histories, as most models fail early if we propagate errors and never reach the genuinely long-horizon parts.
>
> We understand the concern that this is not a "pure" end-to-end online evaluation over the entire task, and we now state this clarifying that the protocol is designed specifically to "stress-test" long-horizon context use.
>
> W4, W5. We have updated the paper to reflect these editorial suggestions
>
> W6. We agree that the phrase “the most practical approach” is too strong given the scope of our experiments and baselines. Our intention was to argue that Interleaved Goal-State-Action modeling is a practical and effective approach under the long-horizon setting we study, rather than to claim absolute optimality over all possible architectures. We also added Memory-Augmented baselines with textual and visual history to provide a more informative comparison.
>
> W7. We appreciate this important nuance. Our work focuses on vision-language-action agents in simulated 3D environments, not the full breadth of embodied AI. We revised the sentence to “capabilities and limitations of vision-language-action agents on long-horizon tasks.”
>
> W8. We rewrote this passage to avoid implying an inherent limitation of multimodal LLMs and instead highlight the mismatch between their typical static training/evaluation regime and our image- and interaction-dominant long-horizon setting.
>
> W9. We clarified that our concern with proprietary API-based models is practical (cost, latency, rate limits, limited control) for large-scale, real-time embodied experiments, not an assumption that they are weaker in capability.
>
> W10, W11. We updated captions so shaded regions in Figure 5 denote variance across test instances, and the y-axis in Figure 4 denotes depth, as in standard needle-in-a-haystack tasks.
>
> W12. We clarified that models in the long-context NiEH evaluation are off-the-shelf to ensure a fair comparison with Gemini (which we cannot fine-tune), and we note that adding fine-tuned results is an avenue for future work.
>
> W13. We agree this qualitative finding is consistent with the broader literature and is not surprising on its own. Our contribution here is to demonstrate and quantify this effect in the embodied, multi-hundred-step tasks, and to do so using a framework that can generate arbitrarily long trajectories and extend context systematically.
>
> W14. We clarified that our architectural contribution is to adapt and systematically study interleaved Goal-State-Action modeling in an extreme long-context embodied setting, combined with our framework and NiEH benchmark, and we now describe this as "exploring" and "studying" architectural considerations rather than introducing a new paradigm.
>
> Q1. We agree that “reasoning” is often overloaded. In our work, we use it in a specific, operational sense.
> For NiEH, this means correctly answering questions that require retrieving and combining information from earlier frames/actions far back in the context; for interactive evaluation, it means choosing the next action based on such long-range dependencies.
>
> Q2. We rephrased this to: “…a new framework for the generation of long-horizon tasks and for the training and evaluation of agents on these tasks.”
>
> Q3. We apologize for the confusion. We’ve updated the new version with appendix
>
> Q6. As noted above, yes: sub-goals are dependent through the shared environment state (object placements, opened containers, etc.). The GT prefixes used to initialize each sub-goal include all previous sub-goals, so the model must operate in a state that reflects the full interaction history.
>
> Q7. We added more detail on data generation (built on AI2-THOR scenes and configurations) and will include a dedicated subsection summarizing the dataset and examples.

---

### Official Review · Reviewer_fpYK · 2025-11-01

**Soundness:** 2
**Presentation:** 2
**Contribution:** 2
**Rating:** 4
**Confidence:** 3

**Summary:**

This paper introduces a new framework aimed at advancing research on long-horizon embodied tasks. The framework provides: (1) infrastructure for generating and evaluating arbitrarily long, reproducible agent trajectories; (2) a benchmark evaluating agents ability to reason over temporally distant multimodal cues; and (3) a dataset with tasks of hundreds of steps and ground-truth actions. The authors explore architectural approaches and training methods to allow agents to handle long contexts.

**Strengths:**

1. The motivation for this work is very solid and timely, as current models for embodied planning and decision making are struggling with long context, often confined to short terms tasks without the ability to perform long horizon optimization. Although the task of recalling details in long horizon action sequences is not directly aiming at the core of the planning problem, it also points at a capability in the right direction.

2. The interleaved Goal–State–Action modeling idea is interesting.

**Weaknesses:**

1. The model claims to explore "ARCHITECTURES FOR LONG-HORIZON VISION-LANGUAGE-ACTION MODELS". However, there is no explicit evaluation of the core task for VLAs: planning. Instead, authors only evaluate the model on the Needles in the Embodied Haystack task of long horizon question answering.

2. The performance gains on this single task does not fully justify the complex architecture changes made. Perhaps one way to more concretely justify the modeling is by experimenting on other tasks more relevant to Embodied Agents, like actual task planning, or established embodiment benchmarks like VSI-Bench, EAI, etc.

3. Assuming that the architecture has its merits, I think this paper has the potential to be a good work, it just needs more time to test different hypothesis to get some more solid results.

**Questions:**

1. Are there any quantitative metrics for the Needles in the Embodied Haystack task, and why are they not reported in the paper?

---

> ### Author Response · Authors · 2025-12-02
>
> We thank you for the valuable and thoughtful feedback. We address the raised limitations below:
>
> 1. We agree that the main objective of VLA models is planning and long-horizon decision making. In addition to the Needles in the Embodied Haystack (NiEH) QA benchmark, our paper does evaluate planning behavior through interactive online evaluation in Section 5.2. We report the results where agents act in the environment to complete long-horizon tasks. These experiments directly test the agent’s ability to plan over hundreds of steps, not just to answer questions. We will revise the text to more explicitly frame Section 5.2 as a planning evaluation and to better connect those results to the architectural choices discussed in Section 4.
>
> 2. We agree that testing on additional embodied benchmarks would further strengthen the empirical case for the architecture. However, our primary goal in this work is to introduce 1) a framework and infrastructure for arbitrarily long and reproducible trajectories, and 2) a long-horizon benchmark (NiEH) specifically designed to stress-test temporally distant multimodal reasoning. The proposed architecture is compatible with other embodied planning tasks, and we see extending our evaluation to those benchmarks as an important next step; we will explicitly mention this in the limitations and future-work section.
>
> For your question, we appreciate the request for clearer quantitative metrics. In the revised version, we have added explicit NiEH accuracy numbers to Table 3 and clarified the definition of the evaluation metric in the main text. We hope this makes the performance on NiEH more transparent and easier to interpret.

---

### Official Review · Reviewer_xoC5 · 2025-11-01

**Soundness:** 2
**Presentation:** 3
**Contribution:** 2
**Rating:** 4
**Confidence:** 2

**Summary:**

The paper tackles long-horizon embodied reasoning agents remembering and acting on events hundreds of steps apart.
IntroducesTHOR, built on AI2-THOR, for generating ultra-long interactive tasks.
Adds Needle(s) in the Embodied Haystack (NiEH) QA tasks where clues are hidden across 600–900+ steps.
Proposes interleaved Goal–State–Action modeling combining vision, language, and action into one token sequence.
Extends transformer context with RoPE scaling, YaRN, and LongRoPE for 100k+ tokens.
Uses Context Parallelism via Ring Attention to train on massive sequences efficiently.
Implements a 7B LLaVA-based agent fine-tuned for long-term reasoning.
Findings: long-memory modeling becomes feasible, but coherence over hundreds of steps remains weak.
Overall: a strong framework and benchmark pushing embodied AI toward true long-term reasoning and planning.

**Strengths:**

Strong problem focus - long-horizon memory in embodied AI, timely and hard.
THOR is scalable and open  can synthesize endless, ultra-long trajectories with full action traces.
NiEH benchmark is unique tests recall of scattered clues across hundreds of steps, bridging vision, language, and long-term reasoning.
Task design is clever enforces early–late dependencies, no shortcuts.
Interleaved Goal–State–Action model clean, unified architecture, handles temporal context elegantly.
Rigorous experiments real ablations on RoPE scaling, YaRN, context length; solid quantitative insight.
Context Parallelism practical and technically hard; shows long-context training is possible, not just theoretical.
Findings are balanced improvement with long context, still breaks past 512k tokens; honest about limits.
Great clarity and visuals figures explain results at a glance.
Open release and reproducibility detailed setup, environment, data planned for release.
Broader impact connects symbolic planning to physical control; builds a base for next-gen long-term embodied agents.

**Weaknesses:**

Relies only on context extension for memory. No exploration of retrieval or hierarchical memory; limits scalability beyond 512k tokens.

No direct baseline against modular vision–language–action models. Claim of superiority for interleaved modeling not empirically proven.

Lacks external dynamics: all environment changes are agent-driven. No tests of memory for unobserved or changing scenes. Models fail beyond 0.5M tokens, multi-evidence QA accuracy drops sharply, and long runs often collapse. Needs clearer absolute metrics.

Statistical rigor missing. No variance, confidence intervals, or multiple seeds reported; unclear if differences are significant. 7B model fine-tuned on 130k tokens with 8×H100s; not practical for most labs. Inference cost not discussed. Planning and manipulation models evaluated separately; no unified control pipeline yet.

**Questions:**

How is the goal given during interaction? Is the final instruction known from the start and repeated each step, or revealed later? Does the agent get any subgoal hints along the way? Clarifying this would help interpret its planning and autonomy.

Did you try other memory idea like retrieval, recurrent state, or hierarchical summaries? Since all RoPE methods fail past 512k tokens, do you think future work should shift toward explicit or learned memory rather than just longer contexts?

Any comparison to a non-interleaved setup, e.g. separate vision + policy modules like ALFRED or PaLM-E? Even a small-scale test would show if the interleaved model truly helps.

What actually breaks beyond 0.5M tokens—GPU memory, instability, or degraded attention? Any thoughts on mixing retrieval or staged training to go further?

Multi-Evidence Reasoning: which question types fail most—temporal, counting, or spatial? Do models usually pick one wrong clue or combine clues incorrectly? A short error breakdown would be great.

If using a larger model (13B–70B), would memory or training cost be the main blocker? Do you expect bigger models to meaningfully extend reasoning, or still hit the same context wall?

---

> ### Author Response · Authors · 2025-12-02
>
> We thank you for the valuable and thoughtful feedback. We address the raised limitations below:
>
> 1. We agree that relying only on context for memory is not the only way to approach long-horizon tasks, and that retrieval or hierarchical memory mechanisms are important next steps. In the updated version, we have added a new baseline: a memory-augmented model that incorporates (i) textual summaries of past environment states and (ii) a visual memory retrieved from past environment states. This baseline uses an explicit history representation instead of simply extending the raw token sequence and provides a more direct comparison with the interleaved Goal-State-Action model.
> We also would like to note that the observed “limit beyond 512k tokens” is a limitation of current LLM backbones and their context-extension methods, not of our framework itself.
>
> 2. In line with the above, we have updated the paper to include a memory-augmented architecture that maintains a representation of previous states as a history, and we compare this to the interleaved Goal-State-Action model.
>
> 3. Our current environments focus on agent-driven dynamics: all scene changes arise from the agent’s actions, we agree this does not cover external dynamics and will clarify this in the limitation.
> Regarding what breaks beyond ~0.5M tokens: for models such as LLaVA-OV and DeepSeek-VL, we observe that performance degrades sharply once the effective context length exceeds what the backbone has seen during pre-training/long-context adaptation. Since GPU memory was managed via Context Parallelism, our conjecture is that this failure is largely due to misalignment in positional encodings at unseen scales (e.g., RoPE scaling beyond the trained regime), rather than purely a hardware limitation.
>
> To make the results more interpretable, we have added explicit accuracy metrics for NiEH (now reported in Table 3). We hope this addresses the request for clearer absolute numbers, beyond visualizations
>
> 4. We appreciate the comment on statistical rigor. Given the high cost of long-context training (7B model, 130k-token sequences, 8×H100), we did not originally report multiple seeds for all configurations. However, we plan to report variance or standard deviation across multiple runs where feasible.
> We also want to clarify that some discrepancies between “planning” and “manipulation” evaluations stem from differences in the underlying simulator interfaces, not from limitations of our framework itself.
>
>
> Here we answer your questions:
>
> Q1. The instruction for the global task (the long-horizon sub-task in Figure 1)  is provided at the beginning of each episode and remains part of the context throughout. The tasks are decomposed into sub-goals during trajectory generation, and the agent receives a sub-goal description at the start of each sub-task phase. Thus, during interaction, the agent observes: 1) global task goal (fixed),  2) current sub-goal instruction, and 3) trajectory prefix (states and actions) up to the current step.
>
> Q2, Q3. As noted above, we agree that retrieval, recurrent state, and hierarchical summaries are important alternatives. Our initial submission focused on the interleaved method with a full context, to isolate long-context scaling effects, but in the revised version we add a memory-augmented baseline that compresses past states into textual and visual summaries and uses these as a history representation.
>
> Q4. Our hypothesis is that the main reason LLaVA-OV and DeepSeek-VL fail beyond 0.5M tokens is positional encoding misalignment: when RoPE (or its variants) is scaled far beyond the context lengths seen during pre-training, the positional embeddings no longer behave reliably, which in turn leads to unstable or ineffective attention.
>
> Q5. In our error analysis, the most frequent failure types for LLaVA-OV and Qwen2.5-VL were temporal reasoning (e.g., confusing the order of events or which object was interacted with earlier vs. later), an object counting (e.g., failing to track pick/place actions). Gemini performs comparatively better on counting questions, but in single-task settings it sometimes hallucinates answers not grounded in the trajectory. We will add a short error breakdown (by question type) in the appendix and summarize the main patterns in the main text.
>
> Q6. Our observations suggest that pre-training and long-context adaptation strategy matter at least as much as model size. For example, Qwen2.5-VL is trained in multiple phases with progressively longer input lengths and additional YaRN-based long-context training, and this correlates with its relatively strong performance at long contexts. This is also consistent with our finding that YaRN works particularly well when aligned with the backbone’s long-context pre-training. We anticipate that larger models (13B–70B) could help, especially for complex reasoning, but they are unlikely to eliminate the context wall on their own without targeted long-context pre-training.

---

### Author Response · Authors · 2025-12-02

We thank the reviewers for the careful reading and constructive feedback. We have revised the paper to address the main concerns and to better highlight the core contribution of this work: a general framework and benchmark for studying long-horizon, VLA agents in embodied environments, not just a single model or task.

Below we summarize the key changes and clarifications.

1. **Additional baselines -- Memory-augmented models**: Beyond the interleaved Goal-State-Action architecture, we now include memory-augmented baselines that maintain compact text-based and image-based summaries of past states. These models explicitly separate long-term history from the immediate token stream. The new results (reported in Table 3) show how these memory variants compare to pure context-extension approaches and help clarify the strengths and limits of interleaved modeling in our setting. We briefly share the results here:

| Model Setting                              | LLaVA-OV | Qwen2.5-VL | Gemini-2.0 Flash |
|-------------------------------------------|:--------:|:----------:|:----------------:|
| (a) Full trajectory (Image-only)          |   0.0    |   44.67    |      35.44       |
| (b) Interleaved Goal-State-Action         |   0.0    |   51.64    |     **83.04**    |
| (c) Memory-Augmented (Text)               |  51.33   |   54.38    |      56.21       |
| (d) Memory-Augmented (Image, Top-10)      |  35.80   |   33.99    |      25.18       |
| (e) Memory-Augmented (Image, Top-20)      |  37.36   |   42.68    |      28.16       |


2. **Clearer dataset construction and interactive evaluation**: We clarified how long trajectories are generated and how online evaluation is performed:
a) Long trajectories are built by composing planner-generated sub-tasks while replaying actions in AI2-THOR, ensuring **continuous and consistent object states** rather than naive concatenation.
b) For interactive evaluation, we now describe in the main text that tasks are decomposed into sub-goals; the agent acts in a closed loop within each sub-goal; and after each sub-goal the context and environment are reset to the planner ground-truth prefix. This design allows us to probe long-horizon behavior even when models fail early, and we explicitly discuss its limitations.

3. **More explicit metrics and error analysis**: We added **NiEH accuracy numbers** and clarified the evaluation metric in the main text (Table 3), rather than relying only on heatmaps. We also provide a brief error breakdown to show which multi-evidence reasoning types are most challenging for current models.

4. **Presentation and figure improvements**: We improved several presentation issues raised by reviewers:
Figure 4 now has explicit axis labels ("Depth" and "Context length"), a color bar indicating NiEH accuracy, and a clearer caption explaining the meaning of white and gray regions.
We introduced PDDL with a proper reference, and moved key details from the appendix into the main text for better readability and reproducibility.

5. **Improved positioning, scope, and claims**: We revised the wording to better match the scope of the work:
We restrict our claims to vision-language-action agents in simulated 3D environments, not all embodied AI.
We soften strong phrases such as “the most practical approach,” emphasizing instead that interleaved Goal-State-Action modeling is a practical and effective design within our experimental setting.
We clarify that limitations such as failures beyond ~512K tokens reflect current backbone/context-extension methods, not constraints of the framework itself.

Overall, our goal is to provide a foundation for long-horizon embodied research: a scalable framework that can synthesize arbitrarily long, state-consistent trajectories; the NiEH benchmark that stresses multi-clue, multi-step reasoning over hundreds of steps; and a systematic analysis of current long-context architectures (and now memory-augmented variants) under these conditions.
Even where some trends (e.g., degradation beyond pre-training context length) may be intuitively expected, we believe our framework, dataset, and empirical characterization offer a concrete, reproducible basis for future work on long-horizon embodied agents.

---

### Meta-Review · Area_Chair_3o3X · 2026-01-06

**Summary:**

The paper proposes the $\infty$-THOR framework and the "Needle(s) in the Embodied Haystack" (NiEH) benchmark to address generation and evaluation of long-horizon embodied tasks. Initially, all four reviewers rated the paper as "4: marginally below acceptance," primarily criticizing the lack of comparison against memory-augmented/retrieval baselines, unclear quantitative metrics (reliance on heatmaps), missing appendices, and lack of clarity on trajectory generation. The authors provided a substantial rebuttal with extensive additional experiments.  However, the PDF uploaded for this submission **reveals the authors’ names and affiliations**, which violates the double-blind review policy.

**Reviewer Concerns:**

Addressed Concerns:

Baselines: The authors added memory-augmented baselines (using text and image summaries), addressing the core concern from Reviewer xoC5 regarding the reliance solely on context extension.

Metrics: Heatmaps were supplemented with explicit accuracy tables (Table 3), resolving clarity issues raised by Reviewers fpYK and rSXS.

Outstanding Concerns:

Interactive Evaluation Protocol: While the authors justified "resetting to Ground Truth context" after sub-goals as a way to stress-test memory without error cascading, this design still limits the assessment of fully autonomous, end-to-end planning capabilities.

**Reviewer Scores:**

Reviewers xoC5 & rSXS: 4 $\rightarrow$ 6 (Key deficits removed).

Reviewers fpYK & fHk7: 4 $\rightarrow$ 5 (Methodological concerns addressed, though some reservations on evaluation scope may remain).

---

### Decision · Program_Chairs · 2026-01-26

Reject